PREPARED FOR SUBMISSION TO JHEP

KA-TP-18-2024
P3H-24-068

# Renormalisation group effects in SMEFT for di-Higgs production

**Gudrun Heinrich,**[a] **Jannis Lang**[a]

[a]*Institute for Theoretical Physics, Karlsruhe Institute of Technology (KIT), Wolfgang-Gaede-Str. 1, 76131 Karlsruhe, Germany*

*E-mail:* gudrun.heinrich@kit.edu, jannis.lang@partner.kit.edu

ABSTRACT: We study the effects of renormalisation group running for the Wilson coefficients of dimension-6 operators contributing to Higgs boson pair production in gluon fusion within Standard Model Effective Field Theory (SMEFT). The running of these Wilson coefficients has been implemented in the NLO QCD code `ggHH_SMEFT`, which is publicly available within the `Powheg-Box-V2` framework.

KEYWORDS: LHC, Higgs-boson couplings, NLO, EFT

# 1 Introduction

The production of Higgs boson pairs is of major importance to elucidate the form of the Higgs potential, at the LHC as well as at future colliders. The form of the Higgs potential assumed in the Standard Model (SM) is reminiscent of an effective potential, yet unknown physics at higher energy scales may alter its form, leading to small deviations from the SM expectation at energies we are currently able to probe.

Parametrising effects of new physics at higher energy scales through Effective Field Theory (EFT) is a meanwhile well established procedure, and the experimental collaborations already have constrained large sets of Wilson coefficients related to an EFT expansion such as Standard Model Effective Field Theory (SMEFT) [1–4] or Higgs Effective Field Theory (non-linear EFT, HEFT) [5–10].

Global fits based on observables from different processes can lift degeneracies in the space of anomalous couplings and may hint to patterns of deviations from the SM. On the other hand, they involve measurements at different energy scales, for example when combining flavour and high energy collider data. Therefore it is important to take the running according to their renormalisation group equations (RGEs) into account. In SMEFT, the one-loop running for dimension-6 operators has been calculated in

Refs. [11–13] and implemented in various tools [14–18]. Work on two-loop running in SMEFT has also started to emerge [19–23]. In HEFT, the one-loop RGEs have been worked out in Refs. [24, 25].

The impact of one-loop RGE running effects of Wilson coefficients on processes relevant for LHC phenomenology, such as Higgs+jet, Higgs pair, $t\bar{t}$ or $t\bar{t}H$ production, has been studied in Refs. [23, 26–30]. However, for Higgs boson pair production in gluon fusion, running Wilson coefficients in combination with full SM NLO QCD corrections, calculated in Refs. [31–34], are not yet available.

For constant Wilson coefficients, the results of [31] have been implemented into the `Powheg-Box-V2` event generator [35–37] allowing for $\kappa_\lambda$ variations in Ref. [38] and including the leading operators contributing to this process for HEFT in Ref. [39, 40] and for SMEFT in Ref. [41]. Recently, the NLO QCD corrections obtained from the combination of a $p_T$-expansion and an expansion in the high-energy regime have been calculated analytically and implemented in the `Powheg-Box-V2` [42]. We note that in HEFT, the QCD renormalisation factorises from the treatment of the leading operators contributing to $gg \to hh$ [25, 43], therefore the running effects are given by the QCD running of the strong coupling (and masses in the $\overline{\text{MS}}$ scheme), when considering only the leading logarithmic (LL) level.

The present work aims to further extend the functionalities of the public code `ggHH_SMEFT` [41], which since recently also includes 4-top-operators and the chromo-magnetic operator [44], providing now the leading RGE running of the operators included in the program. This is a further step on the way to provide tools combining precise predictions in the SM with flexible functionalities for Higgs boson pair production within SMEFT and HEFT.

This paper is structured as follows: in Section 2, the RGE running of the leading and subleading Wilson coefficients contributing to $gg \to hh$ is calculated. Section 3 describes the implementation and usage within the `ggHH_SMEFT` code. In Section 4, we perform phenomenological studies to assess the effects of running Wilson coefficients on Higgs boson pair production, before we conclude in Section 5. The Appendix contains details about power counting in view of the RGEs.

## 2 The RGE running for the relevant subset of Wilson coefficients

We are considering the following operators at dimension-6 in the SMEFT, relevant for Higgs boson pair production in gluon fusion:

$$
\begin{aligned}
\mathcal{L}_{\text{SMEFT}} \supset{} & \mathcal{C}_{H\square}\,(\phi^\dagger\phi)\square(\phi^\dagger\phi) + \mathcal{C}_{HD}\,(\phi^\dagger D_\mu\phi)^*(\phi^\dagger D^\mu\phi) + \mathcal{C}_H\,(\phi^\dagger\phi)^3 \\
& + \mathcal{C}_{tH}\left(\phi^\dagger\phi\bar{Q}_L\tilde{\phi}t_R + \text{h.c.}\right) + \mathcal{C}_{HG}\,\phi^\dagger\phi G^a_{\mu\nu}G^{\mu\nu,a} \\
& + \mathcal{C}_{tG}\left(\bar{Q}_L\sigma^{\mu\nu}T^a G^a_{\mu\nu}\tilde{\phi}t_R + \text{h.c.}\right) \\
& + \mathcal{C}^{(1)}_{Qt}\,\bar{Q}_L\gamma^\mu Q_L\bar{t}_R\gamma_\mu t_R + \mathcal{C}^{(8)}_{Qt}\,\bar{Q}_L\gamma^\mu T^a Q_L\bar{t}_R\gamma_\mu T^a t_R \,,
\end{aligned}
\tag{2.1}
$$

where $\sigma^{\mu\nu} = \frac{i}{2}\left[\gamma^\mu, \gamma^\nu\right]$ and $\tilde{\phi} = i\sigma_2\phi$ is the charge conjugate of the Higgs doublet. The dimensionful Wilson coefficients are also commonly written as $\mathcal{C}_i = \frac{C_i}{\Lambda^2}$ with explicit appearance of the new physics cutoff scale $\Lambda$ to highlight the canonical dimension of a term. The missing 4-top operators with coefficients $\mathcal{C}^{(1)}_{QQ}$, $\mathcal{C}^{(8)}_{QQ}$ and $\mathcal{C}_{tt}$ have been shown to provide no noticeable impact on the cross section of Higgs pair production [44].

The full one-loop RGE of the dimension-6 Wilson coefficients in the Warsaw basis [2] has been derived some time ago and can be found in Refs. [11–13]. According to our operator selection based on power counting arguments and the assumption of weakly coupling UV completions, the first two lines in Eq. (2.1) comprise the leading EFT contribution, while the remaining ones are subleading [44, 45]. Focussing on QCD effects affecting the coefficients in Eq. (2.1), we only need a subset of the one-loop anomalous dimension, namely for $\mathcal{C}_{tH}$, $\mathcal{C}_{HG}$, $\mathcal{C}_{tG}$, $\mathcal{C}^{(1)}_{Qt}$ and $\mathcal{C}^{(8)}_{Qt}$, but have to add the mixing of the 4-top Wilson coefficients into $\mathcal{C}_{HG}$ at two loops. This has been worked out in Ref. [46], where it also has been pointed out that the 4-top-Wilson coefficients depend on the chosen $\gamma_5$ scheme. In this section we restrict the discussion to the Naive Dimensional Regularisation (NDR) scheme [47], yet the RGE solution as implemented in `ggHH_SMEFT` [41, 44] provides a choice between NDR and a version of the Breitenlohner-Maison-t'Hooft-Veltman (BMHV) scheme [48, 49]. Details about the usage are described in Section 3.

The solution of the RGE is represented in analytical form in terms of the scale dependence of the strong coupling $\alpha_s$. To keep a consistent truncation of higher orders, we use the RGE of the strong coupling in the nexto-to-leading-logarithmic (NLL) approximation, which has the form [50]

$$
\mu\frac{\mathrm{d}\alpha_s}{\mathrm{d}\mu} = -2\alpha_s\left(\beta_0\frac{\alpha_s}{4\pi} + \beta_1\frac{\alpha_s^2}{16\pi^2}\right)\,,
\tag{2.2}
$$

with

$$\beta_0 = \frac{11}{3}c_A - \frac{4}{3}T_F n_l \ ,$$
$$\beta_1 = \frac{34}{3}c_A^2 - 4c_F T_F n_l - \frac{20}{3}c_A T_F n_l \ . \tag{2.3}$$

The familiar solution at leading logarithmic (LL) accuracy for the evolution from an input scale $\mu_0$ to $\mu$ is given by

$$\alpha_s^{LL}(\mu) = \frac{\alpha_s(\mu_0)}{1 + \alpha_s(\mu_0)\frac{\beta_0}{2\pi}\log\frac{\mu}{\mu_0}} \ , \tag{2.4}$$

solutions beyond LL are usually provided numerically, e.g. together with the employed parton distribution function (PDF) interfaced to a Monte-Carlo generator.

For later reference, we also list the QCD induced RGE terms of the top-Yukawa coupling in the $\overline{\text{MS}}$ scheme $\overline{y}_t$ [51]

$$\mu\frac{\mathrm{d}\overline{y}_t}{\mathrm{d}\mu} = -2\overline{y}_t\left(\beta_0^y\frac{\alpha_s}{4\pi} + \beta_1^y\frac{\alpha_s^2}{16\pi^2}\right) \ , \tag{2.5}$$

with

$$\beta_0^y = 3c_F \ ,$$
$$\beta_1^y = \frac{3}{2}c_F^2 + \frac{97}{6}c_F c_A - \frac{10}{3}c_F T_F n_l \ , \tag{2.6}$$

and its solution in terms of $\alpha_s$

$$\overline{y}_t^{LL}(\mu) = \overline{y}_t(\mu_0)\left(\frac{\alpha_s^{LL}(\mu)}{\alpha_s(\mu_0)}\right)^{\frac{\beta_0^y}{\beta_0}} \ , \tag{2.7}$$

at LL and

$$\overline{y}_t(\mu) = \overline{y}_t(\mu_0)\left(\frac{\alpha_s(\mu)}{\alpha_s(\mu_0)}\right)^{\frac{\beta_0^y}{\beta_0}}\left(\frac{\beta_0 + \beta_1\frac{\alpha_s(\mu)}{4\pi}}{\beta_0 + \beta_1\frac{\alpha_s(\mu_0)}{4\pi}}\right)^{-\frac{\beta_0^y}{\beta_0} + \frac{\beta_1^y}{\beta_1}} \ , \tag{2.8}$$

at NLL QCD.

## 2.1 SMEFT renormalisation scale dependence and renormalisation group evolution at Leading Log

In general, if we decouple the $\overline{\text{MS}}$ renormalisation scale of the SMEFT coefficients, $\mu_{\text{EFT}}$, from the (QCD) renormalisation scale of the SM, $\mu_R$, the dependence of the cross section on $\mu_{\text{EFT}}$ has two origins. There is an explicit dependence of the fixed order

amplitude on $\mu_{\text{EFT}}$ introduced by the renormalisation of the bare Wilson coefficients, $\mathcal{C}_i^b$, related to the renormalised Wilson coefficients $\mathcal{C}_i$ by

$$\mathcal{C}_i^b = \mu_{\text{EFT}}^{\kappa_i \epsilon} \left( \mathcal{C}_i + \delta_{\mathcal{C}_i}^{\mathcal{C}_j} \mathcal{C}_j \right) , \qquad (2.9)$$

where $\kappa_i$ is a natural number determined by the field (and SM coupling) content of the operator. For convenience, we summarise the relevant counter terms $\delta_{\mathcal{C}_i}^{\mathcal{C}_j}$ entering our calculation of $gg \to hh$:

$$
\begin{aligned}
\delta_{\mathcal{C}_{tH}}^{\mathcal{C}_{tH}} &= \frac{(4\pi e^{-\gamma_E})^\epsilon}{\epsilon} \frac{\alpha_s}{4\pi} \left( -\beta_0^y \left( \frac{\mu_R^2}{\mu_{\text{EFT}}^2} \right)^\epsilon \right) , \\
\delta_{\mathcal{C}_{HG}}^{\mathcal{C}_{HG}} &= \frac{(4\pi e^{-\gamma_E})^\epsilon}{\epsilon} \frac{\alpha_s}{4\pi} \left( -\beta_0 \left( \frac{\mu_R^2}{\mu_{\text{EFT}}^2} \right)^\epsilon + \left( \frac{\mu_R^2}{m_t^2} \right)^\epsilon \frac{4}{3} T_F \right) , \\
\delta_{\mathcal{C}_{tH}}^{\mathcal{C}_{Qt}^{(1)}} &= \frac{(4\pi e^{-\gamma_E})^\epsilon}{\epsilon} \frac{1}{16\pi^2} \frac{\gamma_{\mathcal{C}_{tH}}^{\mathcal{C}_{Qt}}}{2} \left( \frac{\mu_R^2}{\mu_{\text{EFT}}^2} \right)^\epsilon , \\
\delta_{\mathcal{C}_{tH}}^{\mathcal{C}_{Qt}^{(8)}} &= c_F \delta_{\mathcal{C}_{tH}}^{\mathcal{C}_{Qt}^{(1)}} , \\
\delta_{\mathcal{C}_{HG}}^{\mathcal{C}_{tG}} &= \frac{(4\pi e^{-\gamma_E})^\epsilon}{\epsilon} \frac{g_s}{16\pi^2} \frac{\gamma_{\mathcal{C}_{HG}}^{\mathcal{C}_{tG}}}{2} \left( \frac{\mu_R^2}{\mu_{\text{EFT}}^2} \right)^\epsilon , \\
\delta_{\mathcal{C}_{HG}}^{\mathcal{C}_{Qt}^{(1)}} &= \frac{(4\pi e^{-\gamma_E})^{2\epsilon}}{\epsilon} \frac{g_s}{16\pi^2} \frac{\gamma_{\mathcal{C}_{HG}}^{\mathcal{C}_{Qt}}}{4} \left( \frac{\mu_R^2}{\mu_{\text{EFT}}^2} \right)^{2\epsilon} , \\
\delta_{\mathcal{C}_{HG}}^{\mathcal{C}_{Qt}^{(8)}} &= (c_F - \frac{c_A}{2}) \delta_{\mathcal{C}_{tH}}^{\mathcal{C}_{Qt}^{(1)}} ,
\end{aligned}
\qquad (2.10)
$$

where we introduced

$$
\begin{aligned}
\gamma_{\mathcal{C}_{tH}}^{\mathcal{C}_{Qt}} &= 8y_t \left( y_t^2 - \lambda \right) = \frac{4\sqrt{2} m_t \left( 4m_t^2 - m_h^2 \right)}{v^3} , \\
\gamma_{\mathcal{C}_{HG}}^{\mathcal{C}_{tG}} &= 8T_F y_t = 8\sqrt{2} T_F \frac{m_t}{v} , \\
\gamma_{\mathcal{C}_{HG}}^{\mathcal{C}_{Qt}} &= -8T_F y_t^2 \frac{1}{16\pi^2} = -16 T_F \frac{m_t^2}{v^2} \frac{1}{16\pi^2} .
\end{aligned}
\qquad (2.11)
$$

On top of the explicit dependence on $\mu_{\text{EFT}}$ in terms of fixed order renormalisation, the cross section also implicitly depends on $\mu_{\text{EFT}}$ due to the scaling of the renormalised Wilson coefficients dictated by the RGE. Our selection of relevant RGE terms follows from a few guiding principles:

We are interested only in the Wilson coefficients listed in Eq. (2.1), as these lead to non-negligible contributions to the cross section without running effects following the investigation in Ref. [44], and consider only QCD effects at LL plus those non-QCD

terms which are induced by the renormalisation necessary for a finite contribution of the subleading Wilson coefficients to $gg \to hh$ listed in Eq. (2.10).

Moreover, since we want to describe the scaling behaviour relevant for a precise measurement of Wilson coefficients at current collider energies and not incorporate the running up to or down from an arbitrarily high energy scale, we employ a decoupling of heavy particles like the top-quark and the Higgs boson from the RGE, as their logarithmic contributions should not be of high impact. This also has to be respected in the NLO QCD calculation, which is implicitly considered by the counter term $\delta_{\mathcal{C}_{HG}}^{\mathcal{C}_{HG}}$ in Eq. (2.10), implying a decoupling of the top-quark at $m_t$. In addition, we fix the mass of the top-quark (and associated Yukawa-parameter) with an on-shell renormalisation. A subsequent running between measurement scale and UV matching scale can be performed independently with all SM particles and parameters being active[1], and may include higher logarithmic accuracy, as the large scale difference might require much more precision.

Following these criteria, the remaining terms of the RGEs are given by

$$
\mu \frac{\mathrm{d}\mathcal{C}_{tH}}{\mathrm{d}\mu} = -2\beta_0^y \frac{\alpha_s}{4\pi} \mathcal{C}_{tH} + \gamma_{\mathcal{C}_{tH}}^{\mathcal{C}_{Qt}} \frac{1}{16\pi^2} \left( \mathcal{C}_{Qt}^{(1)} + c_F \mathcal{C}_{Qt}^{(8)} \right) \ ,
$$

$$
\mu \frac{\mathrm{d}\mathcal{C}_{HG}}{\mathrm{d}\mu} = -2\beta_0 \frac{\alpha_s}{4\pi} \mathcal{C}_{HG} + \gamma_{\mathcal{C}_{HG}}^{\mathcal{C}_{tG}} \frac{g_s}{16\pi^2} \mathcal{C}_{tG} + \gamma_{\mathcal{C}_{HG}}^{\mathcal{C}_{Qt}} \frac{\alpha_s}{4\pi} \left( \mathcal{C}_{Qt}^{(1)} + (c_F - \frac{c_A}{2})\mathcal{C}_{Qt}^{(8)} \right) \ ,
$$

$$
\mu \frac{\mathrm{d}\mathcal{C}_{tG}}{\mathrm{d}\mu} = -\beta_{tG} \frac{\alpha_s}{4\pi} \mathcal{C}_{tG} \ , \tag{2.12}
$$

$$
\begin{pmatrix} \mu \frac{\mathrm{d}\mathcal{C}_{Qt}^{(1)}}{\mathrm{d}\mu} \\ \mu \frac{\mathrm{d}\mathcal{C}_{Qt}^{(8)}}{\mathrm{d}\mu} \end{pmatrix} = -\hat{\beta}_{Qt} \frac{\alpha_s}{4\pi} \begin{pmatrix} \mathcal{C}_{Qt}^{(1)} \\ \mathcal{C}_{Qt}^{(8)} \end{pmatrix} \ ,
$$

with diagonal coefficients

$$
\beta_{tG} = \beta_0 + 4c_A - 10c_F \ ,
$$

$$
\hat{\beta}_{Qt} = \begin{pmatrix} 0 & 3\frac{N_c^2-1}{N_c^2} \\ 12 & \frac{6N_c^2-2N_c-12}{N_c} \end{pmatrix} \ . \tag{2.13}
$$

Let us comment on how we applied the selection criteria to arrive at the expressions above.

---

[1] For $\mu_0 = m_t$ the decoupling of the top quark is continuous at the considered order in perturbation theory, such that the 5-flavour version of $\mathcal{C}_{HG}$ coincides with the 6-flavour version at the measurement scale. This is desireable, as the running between measurement and matching requires the 6-flavour version which would be directly available in that case. For choices of $\mu_0$ in the vicinity of $m_t$, the discrepancy neglecting the scale difference is numerically irrelevant, such that the identification of the 5-flavour and 6-flavour version at $\mu_0$ is still possible. Even for $\mu_0 = 1$ TeV the effect of the associated mismatch on the total cross section is less than $2\,\%$ of the Born cross section.

- Identifying the QCD terms contained in the results of Ref. [13] entering at LL relies on the application of a power counting scheme for which we provide some details in Appendix A.

- The coefficients $\mathcal{C}_{Qt}^{(1)}$ and $\mathcal{C}_{Qt}^{(8)}$ are not closed under QCD corrections, as the four-fermion Wilson coefficients mix heavily with each other under renormalisation [13]. Neglecting these additional mixings is expected to lead to less than 10% effects on the contribution of $\mathcal{C}_{Qt}^{(1)}$ and $\mathcal{C}_{Qt}^{(8)}$ to the cross section. This has been estimated by investigating the effect of $\mathcal{C}_{Qt}^{(1)}$ or $\mathcal{C}_{Qt}^{(8)}$ defined at the input scale $\mu_0 = 1$ TeV on the cross section when comparing the full RGE solution with a version with constant subleading coefficient. For realistic scenarios within ranges presented in Fig. 4, the missing contribution is expected to be well within the scale uncertainty range.

- Relying on power counting arguments alone, there would be an additional mixing of $\mathcal{C}_{HG}$ into $\mathcal{C}_{tG}$ in Eq. (2.12) as part of the LL QCD effects, cf. Appendix A. Yet, since the one-particle-irreducible diagrams generating this mixing require a Higgs propagator, this mixing term should be decoupled according to our selection criteria. Moreover, this mixing is also irrelevant from the numerical point of view, as the measured limits of $\mathcal{C}_{HG}$ are much tighter than the bounds on $\mathcal{C}_{tG}$ itself [52] and the effect of the subleading Wilson coefficient $\mathcal{C}_{tG}$ is numerically much smaller than the dependence on the leading contribution of $\mathcal{C}_{HG}$.

In order to solve Eq. (2.12) in terms of $\alpha_s$ we first find the solution to the homogeneous terms described by $\beta_i$ and subsequently add the inhomogeneous solution by the introduction of scale dependent coefficients. We thus find for the running from the input scale of the SMEFT coefficients, $\mu_0$, to $\mu_{\mathrm{EFT}}$

$$
\begin{aligned}
\mathcal{C}_{tH}^{LL}(\mu_{\mathrm{EFT}}) &= \left(\frac{\alpha_s(\mu_{\mathrm{EFT}})}{\alpha_s(\mu_0)}\right)^{\frac{\beta_0^y}{\beta_0}} \left(\mathcal{C}_{tH}(\mu_0) + \Delta\mathcal{C}_{tH}(\mu_{\mathrm{EFT}}, \mathcal{C}_{Qt}^{(1)}, \mathcal{C}_{Qt}^{(8)})\right) , \\
\mathcal{C}_{HG}^{LL}(\mu_{\mathrm{EFT}}) &= \frac{\alpha_s(\mu_{\mathrm{EFT}})}{\alpha_s(\mu_0)} \left(\mathcal{C}_{HG}(\mu_0) + \Delta\mathcal{C}_{HG}(\mu_{\mathrm{EFT}}, \mathcal{C}_{tG}, \mathcal{C}_{Qt}^{(1)}, \mathcal{C}_{Qt}^{(8)})\right) , \\
\mathcal{C}_{tG}(\mu_{\mathrm{EFT}}) &= \left(\frac{\alpha_s(\mu_{\mathrm{EFT}})}{\alpha_s(\mu_0)}\right)^{\frac{\beta_{tG}}{2\beta_0}} \mathcal{C}_{tG}(\mu_0) , \\
\begin{pmatrix} \mathcal{C}_{Qt}^{(1)}(\mu_{\mathrm{EFT}}) \\ \mathcal{C}_{Qt}^{(8)}(\mu_{\mathrm{EFT}}) \end{pmatrix} &= \exp\left(\log\left(\frac{\alpha_s(\mu_{\mathrm{EFT}})}{\alpha_s(\mu_0)}\right) \hat{\beta}_{Qt}\frac{1}{2\beta_0}\right) \begin{pmatrix} \mathcal{C}_{Qt}^{(1)}(\mu_0) \\ \mathcal{C}_{Qt}^{(8)}(\mu_0) \end{pmatrix} .
\end{aligned}
\tag{2.14}
$$

The formal solution of $\left(\mathcal{C}_{Qt}^{(1)}(\mu_{\mathrm{EFT}}), \mathcal{C}_{Qt}^{(8)}(\mu_{\mathrm{EFT}})\right)^T$ is impractical for the determination of $\Delta\mathcal{C}_{tH}(\mu_{\mathrm{EFT}}, \mathcal{C}_{Qt}^{(1)}, \mathcal{C}_{Qt}^{(8)})$ and $\Delta\mathcal{C}_{HG}(\mu_{\mathrm{EFT}}, \mathcal{C}_{tG}, \mathcal{C}_{Qt}^{(1)}, \mathcal{C}_{Qt}^{(8)})$, we therefore express the result

in terms of the diagonalised system of the last equation in Eq. (2.12). We introduce the rotation matrix $R_{Qt}$, which leads to a diagonalisation of[2]

$$R_{Qt}\hat{\beta}_{Qt}R_{Qt}^{-1} = \begin{pmatrix} \beta_{Qt,1} & 0 \\ 0 & \beta_{Qt,2} \end{pmatrix} =: \text{diag}\left(\beta_{Qt,i}\right) , \tag{2.15}$$

with eigenvalues

$$\beta_{Qt,1/2} = \frac{1}{2}\left((\hat{\beta}_{Qt})_{88} \mp \sqrt{(\hat{\beta}_{Qt})_{88}^2 - 4(\hat{\beta}_{Qt})_{18}(\hat{\beta}_{Qt})_{81}}\right) . \tag{2.16}$$

With this definition, we may write

$$\begin{pmatrix} \mathcal{C}_{Qt}^{(1)}(\mu_{\text{EFT}}) \\ \mathcal{C}_{Qt}^{(8)}(\mu_{\text{EFT}}) \end{pmatrix} = R_{Qt}^{-1}\,\text{diag}\left(\left(\frac{\alpha_s(\mu_{\text{EFT}})}{\alpha_s(\mu_0)}\right)^{\frac{\beta_{Qt,i}}{2\beta_0}}\right)R_{Qt}\begin{pmatrix} \mathcal{C}_{Qt}^{(1)}(\mu_0) \\ \mathcal{C}_{Qt}^{(8)}(\mu_0) \end{pmatrix} . \tag{2.17}$$

For the full LL expression of $\mathcal{C}_{tH}$ and $\mathcal{C}_{HG}$ we finally find

$$\mathcal{C}_{tH}^{LL}(\mu_{\text{EFT}}) = \left(\frac{\alpha_s(\mu_{\text{EFT}})}{\alpha_s(\mu_0)}\right)^{\frac{\beta_0^y}{\beta_0}}\left\{\mathcal{C}_{tH}(\mu_0)\right.$$
$$\left. + \frac{\gamma_{\mathcal{C}_{tH}}^{\mathcal{C}_{Qt}}}{g_s^2(\mu_0)}\left(1,\,c_F\right)R_{Qt}^{-1}\,\text{diag}\left(\frac{1-\left(\frac{\alpha_s(\mu_{\text{EFT}})}{\alpha_s(\mu_0)}\right)^{\frac{\beta_{Qt,i}-2\beta_0^y-2\beta_0}{2\beta_0}}}{\beta_{Qt,i}-2\beta_0^y-2\beta_0}\right)R_{Qt}\begin{pmatrix} \mathcal{C}_{Qt}^{(1)}(\mu_0) \\ \mathcal{C}_{Qt}^{(8)}(\mu_0) \end{pmatrix}\right\} , \tag{2.18}$$

and

$$\mathcal{C}_{HG}^{LL}(\mu_{\text{EFT}}) = \frac{\alpha_s(\mu_{\text{EFT}})}{\alpha_s(\mu_0)}\left\{\mathcal{C}_{HG}(\mu_0) + \frac{\gamma_{\mathcal{C}_{HG}}^{\mathcal{C}_{tG}}}{g_s(\mu_0)}\frac{1-\left(\frac{\alpha_s(\mu_{\text{EFT}})}{\alpha_s(\mu_0)}\right)^{\frac{\beta_{tG}-3\beta_0}{2\beta_0}}}{\beta_{tG}-3\beta_0}\mathcal{C}_{tG}(\mu_0)\right.$$
$$\left. + \gamma_{\mathcal{C}_{HG}}^{\mathcal{C}_{Qt}}\left(1,\,c_F-\frac{c_A}{2}\right)R_{Qt}^{-1}\,\text{diag}\left(\frac{1-\left(\frac{\alpha_s(\mu_{\text{EFT}})}{\alpha_s(\mu_0)}\right)^{\frac{\beta_{Qt,i}-2\beta_0}{2\beta_0}}}{\beta_{Qt,i}-2\beta_0}\right)R_{Qt}\begin{pmatrix} \mathcal{C}_{Qt}^{(1)}(\mu_0) \\ \mathcal{C}_{Qt}^{(8)}(\mu_0) \end{pmatrix}\right\} . \tag{2.19}$$

## 2.2 Inclusion of QCD effects beyond Leading Log

Since the fixed order calculation includes NLO QCD corrections to the leading SMEFT contribution, it is desireable to include QCD effects at the next-to-leading-logarithmic

---

[2]The definition on the right of Eq. (2.15) serves as a short-hand notation we use for 2-dimensional diagonal matrices, where the diagonal entries are evaluated with $i = 1, 2$.

(NLL) level as well. As the full NLL scaling of the SMEFT is yet unknown, we follow the procedure of Ref. [27] and include an overall factor multipying the LL solutions of Eqs. (2.18) and (2.19), which describes the NLL QCD evolution of the respective homogeneous part of the RGE. The additional RGE term for $\mathcal{C}_{HG}$ can be derived from Eq. (19) of Ref. [27] identifying $c_g = \frac{\mathcal{C}_{HG}}{\alpha_s}$. Since $\mathcal{O}_{tH}$ has the same structure as the Yukawa interaction regarding QCD corrections, the additional term for $\mathcal{C}_{tH}$ follows from Eq. (2.5). The final version of the RGE has the form

$$\mu\frac{\mathrm{d}\mathcal{C}_{tH}}{\mathrm{d}\mu} = -2\left(\beta_0^y\frac{\alpha_s}{4\pi} + \beta_1^y\left(\frac{\alpha_s}{4\pi}\right)^2\right)\mathcal{C}_{tH} + \gamma_{\mathcal{C}_{tH}}^{\mathcal{C}_{Qt}}\frac{1}{16\pi^2}\left(\mathcal{C}_{Qt}^{(1)} + c_F\mathcal{C}_{Qt}^{(8)}\right) \,,$$

$$\mu\frac{\mathrm{d}\mathcal{C}_{HG}}{\mathrm{d}\mu} = -2\left(\beta_0\frac{\alpha_s}{4\pi} + 2\beta_1\left(\frac{\alpha_s}{4\pi}\right)^2\right)\mathcal{C}_{HG} + \gamma_{\mathcal{C}_{HG}}^{\mathcal{C}_{tG}}\frac{g_s}{16\pi^2}\mathcal{C}_{tG} + \gamma_{\mathcal{C}_{HG}}^{\mathcal{C}_{Qt}}\frac{\alpha_s}{4\pi}\left(\mathcal{C}_{Qt}^{(1)} + (c_F - \frac{c_A}{2})\mathcal{C}_{Qt}^{(8)}\right) \,,$$

$$\mu\frac{\mathrm{d}\mathcal{C}_{tG}}{\mathrm{d}\mu} = -\beta_{tG}\frac{\alpha_s}{4\pi}\mathcal{C}_{tG} \,,$$

$$\begin{pmatrix}\mu\frac{\mathrm{d}\mathcal{C}_{Qt}^{(1)}}{\mathrm{d}\mu} \\ \mu\frac{\mathrm{d}\mathcal{C}_{Qt}^{(8)}}{\mathrm{d}\mu}\end{pmatrix} = -\hat{\beta}_{Qt}\frac{\alpha_s}{4\pi}\begin{pmatrix}\mathcal{C}_{Qt}^{(1)} \\ \mathcal{C}_{Qt}^{(8)}\end{pmatrix} \,,$$

$$(2.20)$$

and we have approximate solutions

$$\mathcal{C}_{tH}(\mu_{\mathrm{EFT}}) = \mathcal{C}_{tH}^{LL}(\mu_{\mathrm{EFT}})\left(\frac{\beta_0 + \beta_1\frac{\alpha_s(\mu_{\mathrm{EFT}})}{4\pi}}{\beta_0 + \beta_1\frac{\alpha_s(\mu_0)}{4\pi}}\right)^{-\frac{\beta_0^y}{\beta_0} + \frac{\beta_1^y}{\beta_1}} \,,$$

$$\mathcal{C}_{HG}(\mu_{\mathrm{EFT}}) = \mathcal{C}_{HG}^{LL}(\mu_{\mathrm{EFT}})\left(\frac{\beta_0 + \beta_1\frac{\alpha_s(\mu_{\mathrm{EFT}})}{4\pi}}{\beta_0 + \beta_1\frac{\alpha_s(\mu_0)}{4\pi}}\right) \,.$$

$$(2.21)$$

# 3 Implementation and usage within the `Powheg-Box-V2`

The RGE evolution described in the previous section is implemented as an extension of `ggHH_SMEFT` [41, 44] while the usage of the preexisting features is the same. We added some new options in the input card which define the RGE flow of the coefficients:

`WCscaledependence`: Switches the scale dependence of the Wilson Coefficients between three modes,

0: $\mu_{\mathrm{EFT}} = \mu_R$ but without any running effects (default, represents previous implementation)

1: static EFT scale $\mu_{\mathrm{EFT}} = \mu_0$.

Takes the fixed value for the EFT input scale defined by the user. If `EFTscfact` $\neq 1$, $\mu_{\mathrm{EFT}} = \mu_0 \times$ `EFTscfact`, such that running

between $\mu_0$ and $\mu_{\mathrm{EFT}}$ is included.

2: dynamic EFT scale, $\mu_{\mathrm{EFT}} = \frac{m_{hh}}{2} \times$ `EFTscfact` with running.

`inputscaleEFT`: defines the input scale/measurement scale $\mu_0$ of the Wilson coefficients, from which the running starts.
(Only relevant for `WCscaledependence` $> 0$.)

`EFTscfact`: varies the EFT scale $\mu_{\mathrm{EFT}}$ around the central EFT scale, i.e. $\mu_{\mathrm{EFT}} = \mu_{EFT_{\mathrm{central}}} \times$ `EFTscfact`, to be used for uncertainty assessment.

The new features require the code to be run in SMEFT mode, i.e. setting `usesmeft` $= 1$, since it describes the running of the SMEFT Wilson coefficients. In addition, `includesubleading` $= 0$ [44] switches off the mixing terms of the subleading coefficients into the leading coefficients. No additional requirement is put on the setting of `SMEFTtruncation` (this keyword supersedes `multiple-insertion`),[3] nevertheless only `SMEFTtruncation` $= 0, 1$ corresponding to truncation options (a) (SM+linear EFT) and (b) (SM+linear and quadratic EFT) [41], respectively, are valid choices in a consistent SMEFT power counting.

As a reminder, these truncation options correspond to

$$\mathrm{d}\sigma = \underbrace{\underbrace{\mathrm{d}\sigma_{\mathrm{SM}} + \mathrm{d}\sigma_{\mathrm{dim6}}}_{(a)} + \mathrm{d}\sigma_{\mathrm{dim6}\times\mathrm{dim6}}}_{(b)} , \tag{3.1}$$

where

$$
\begin{aligned}
\mathrm{d}\sigma_{\mathrm{dim6}} &:= \sum_i \mathcal{C}_i(\mu_{\mathrm{EFT}})\mathrm{d}\sigma_i\left(\mu_R, \mu_F, \mu_{\mathrm{EFT}}\right) & &\sim \sum_i \mathcal{C}_i \Re\left(\mathcal{M}_{\mathrm{SM}}\mathcal{M}_i^*\right) , \\
\mathrm{d}\sigma_{\mathrm{dim6}\times\mathrm{dim6}} &:= \sum_{i,j} \mathcal{C}_i(\mu_{\mathrm{EFT}})\mathcal{C}_j(\mu_{\mathrm{EFT}})\mathrm{d}\sigma_{i\times j}\left(\mu_R, \mu_F, \mu_{\mathrm{EFT}}\right) & &\sim \sum_{i,j} \mathcal{C}_i\mathcal{C}_j \Re\left(\mathcal{M}_i\mathcal{M}_j^*\right) ,
\end{aligned}
\tag{3.2}
$$

denote the linear interference with the SM and quadratic contribution to the cross section, respectively. The sums over $i$, $j$ include all Wilson coefficients of Eq. (2.1). In addition, the amplitudes are further expanded according to perturbation theory in the SM couplings; we provide NLO QCD corrections to the SM and the leading Wilson

---

[3]`multiple-insertion` is still available for backwards compatibility, however the setting of `SMEFTtruncation` overrules the value given for `multiple-insertion`.

coefficient contributions involving $\{\mathcal{C}_{H\square}, \mathcal{C}_{HD}, \mathcal{C}_H, \mathcal{C}_{tH}, \mathcal{C}_{HG}\}$. This leads to

$$
\begin{aligned}
\mathrm{d}\sigma_{\mathrm{SM}} &= \mathrm{d}\sigma_{\mathrm{SM}}^{\mathrm{LO}} + \mathrm{d}\sigma_{\mathrm{SM}}^{\mathrm{NLO}} \\
\mathrm{d}\sigma_i &= \begin{cases} \mathrm{d}\sigma_i^{\mathrm{LO}} + \mathrm{d}\sigma_i^{\mathrm{NLO}} & \text{for } i \in \{\mathcal{C}_{H\square}, \mathcal{C}_{HD}, \mathcal{C}_H, \mathcal{C}_{tH}, \mathcal{C}_{HG}\}\,, \\ \mathrm{d}\sigma_i^{\mathrm{LO}} & \text{for } \mathcal{C}_{tG}, 4\text{-top operators}\,, \end{cases} \\
\mathrm{d}\sigma_{i\times j} &= \begin{cases} \mathrm{d}\sigma_{i\times j}^{\mathrm{LO}} + \mathrm{d}\sigma_{i\times j}^{\mathrm{NLO}} & \text{for } i,j \in \{\mathcal{C}_{H\square}, \mathcal{C}_{HD}, \mathcal{C}_H, \mathcal{C}_{tH}, \mathcal{C}_{HG}\}\,, \\ \mathrm{d}\sigma_{i\times j}^{\mathrm{LO}} & \text{for } \mathcal{C}_{tG}, 4\text{-top operators}\,. \end{cases}
\end{aligned}
\tag{3.3}
$$

In the following section, we refer to this expansion when discussing the inclusion of NLO QCD effects.

## 4 Results

The results presented in this section were obtained for a centre-of-mass energy of $\sqrt{s} = 13.6\,\mathrm{TeV}$ using the PDF4LHC15_nlo_30_pdfas [53] parton distribution functions, interfaced to our code via LHAPDF [54], along with the corresponding value for $\alpha_s$. We used $m_h = 125\,\mathrm{GeV}$ for the mass of the Higgs boson; the top quark mass has been fixed to $m_t = 173\,\mathrm{GeV}$ to be coherent with the virtual two-loop amplitude calculated numerically, the top quark and Higgs boson widths have been set to zero. We set the central renormalisation and factorisation scales to $\mu_R = \mu_F = m_{hh}/2$.

For the reference distributions representing the previously existing implementation we set $\mu_{\mathrm{EFT}} = \mu_R$, but keep the values of the Wilson coefficients fixed, and use a 3-point scale variation of $\mu_R = \mu_F = m_{hh}/2 \cdot \{1, 2, \frac{1}{2}\}$. The central SMEFT scale for the other distributions is either chosen to be static with $\mu_{\mathrm{EFT}} = \mu_0$ ($\mu_{\mathrm{EFT}}$ fixed) or dynamic with $\mu_{\mathrm{EFT}} = m_{hh}/2$ ($\mu_{\mathrm{EFT}}$ dynamic) and the scale variation is only applied to $\mu_{\mathrm{EFT}}$.

We start by considering the behaviour of a single Wilson coefficient, with $\mathcal{C}_i(\mu_0) = 1\,\mathrm{TeV}^{-2}$ as input, and focus only on the interference term ($\mathrm{d}\sigma_{\mathrm{dim6}}$), defined in Eq. (3.2). We compare five different settings for the RGE effects: $\mu_{\mathrm{EFT}} = \mu_R$ with constant coefficients (reference distribution, denoted by "without $\mu_{\mathrm{EFT}}$ dependence"), and both, fixed and dynamic $\mu_{\mathrm{EFT}}$, for input scales of $\mu_0 = 200\,\mathrm{GeV}$ and $\mu_0 = 1\,\mathrm{TeV}$.

Fig. 1 demonstrates the running effects for the leading coefficients $\mathcal{C}_{tH}$ and $\mathcal{C}_{HG}$ individually, as these coefficients only contribute to the diagonal part of the anomalous dimension. The upper panels only show the non-vanishing values of the Wilson coefficients for each scenario, the middle panels show the distributions at LO and the lower panels at NLO QCD. We observe that distributions with $\mu_0 = 200\,\mathrm{GeV}$ are compatible with the reference distributions where no $\mu_{\mathrm{EFT}}$ dependence was included. Including $\mu_{\mathrm{EFT}}$ and using $\mu_0 = 1\,\mathrm{TeV}$, the running effects for $\mathcal{C}_{tH}$ are within the SM 3-point

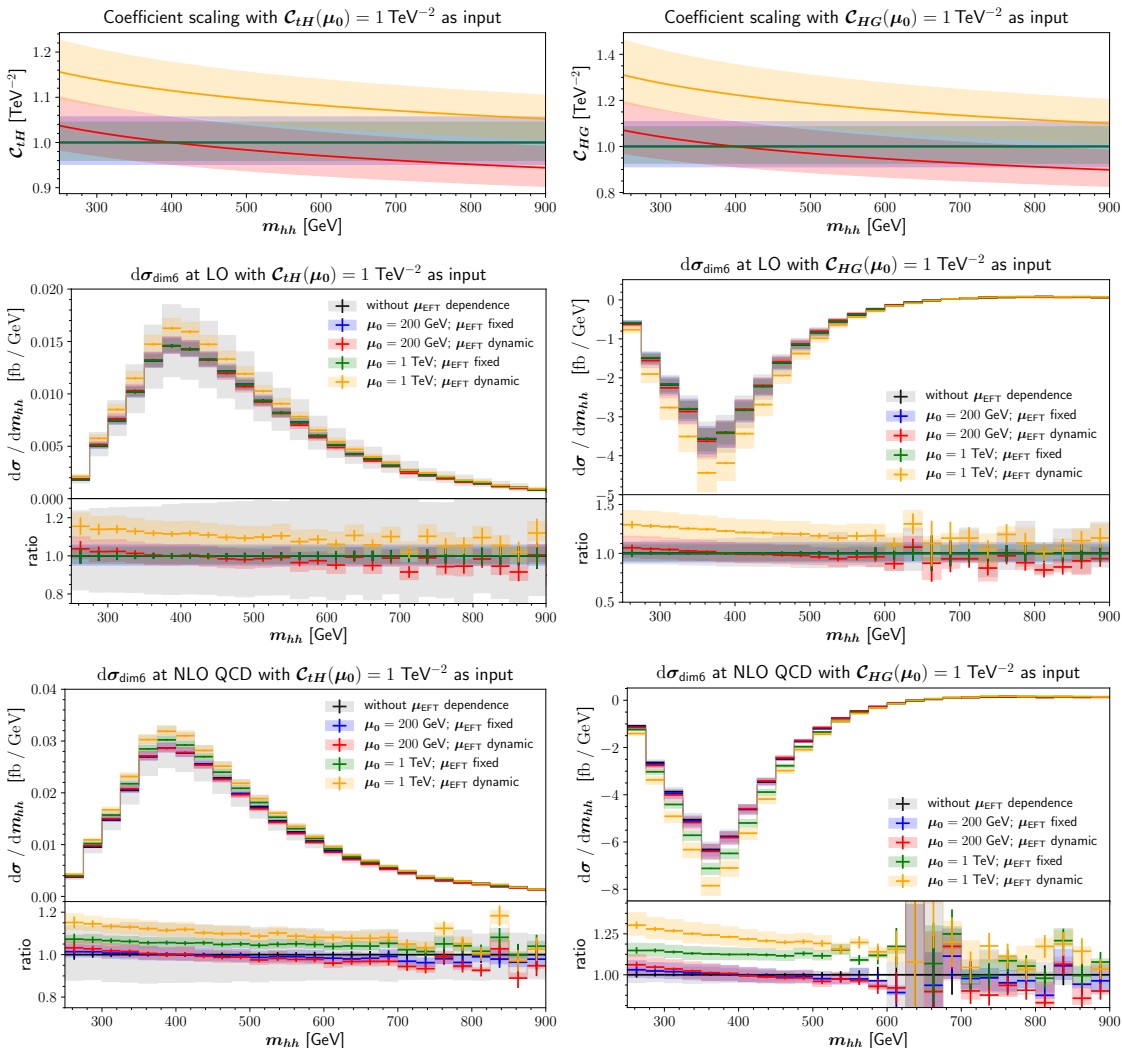

**Figure 1**: Top row: Running of the Wilson coefficients as a function of $m_{hh}$, middle and bottom row: $m_{hh}$ differential cross sections for $\mathrm{d}\sigma_{\mathrm{dim6}}$ at LO and NLO QCD, respectively, with different modes of EFT running, considering only the contribution of a single Wilson coefficient. Left: $\mathcal{C}_{tH}(\mu_0) = 1\,\mathrm{TeV}^{-2}$ as input, right: $\mathcal{C}_{HG}(\mu_0) = 1\,\mathrm{TeV}^{-2}$ as input; upper: values of the Wilson coefficients. The grey bands denote 3-point scale variations of $\mu_R$ and $\mu_F$, while the coloured bands denote variations of $\mu_{\mathrm{EFT}}$ by a factor of two around its central value.

scale uncertainty band (the grey band in Fig. 1), whereas for $\mathcal{C}_{HG}$ the running has a noticeable effect beyond the SM scale uncertainty. However, this may originate from the fact that there is a factor of $\alpha_s$ less in the amplitude, since we do not factor out $g_s$

explicitly from the Wilson coefficient, sticking to the Warsaw basis conventions. This reduces the SM scale uncertainty, but adds an additional contribution to the running of $\mathcal{C}_{HG}$, thus leading to a more significant dependence on the variation of $\mu_{\text{EFT}}$. This fact emphasizes the necessity to estimate the scale uncertainties in a combined manner, to account for different conventions concerning the absorption of SM couplings into Wilson coefficients.

We note that, by construction, the central scale prediction for an input scale $\mu_0 = 200$ GeV precisely coincides with the reference distribution at $m_{hh} = 400$ GeV, since $\mu_R = \mu_F = m_{hh}/2$.

In Fig. 2 we investigate the running for the input $\mathcal{C}_{tG}(\mu_0) = 1$ TeV$^{-2}$. RGE running where only $\mathcal{C}_{tG}$ is non-zero at the input scale $\mu_0$ implies a mixing into $\mathcal{C}_{HG}$, which is induced due to the off-diagonal terms in the anomalous dimension in Eq. (2.12); the scale dependent values of the two Wilson coefficients are depicted in the upper panel. Comparing the distributions in the lower panels, we notice that the two different choices for the EFT input scale lead to vastly different shapes. While $\mu_0 = 200$ GeV is within the scale uncertainty of the reference distribution, choosing $\mu_0 = 1$ TeV leads to results with a different sign for the interference contribution $d\sigma_{\text{dim6}}$. These differences clearly demonstrate that the specification of the input scale is necessary to obtain non-ambiguous results. We also observe that the effects of the running, especially the uncertainties due to variations of $\mu_{\text{EFT}}$, are more pronounced when NLO corrections are included. This can be understood by remembering how the Wilson coefficients enter at the different orders in QCD (see Eq. (3.3)): the contributions multiplying $\mathcal{C}_{tG}(\mu_{\text{EFT}})$ (and also $\mathcal{C}_{Qt}^{(1)}(\mu_{\text{EFT}})$ or $\mathcal{C}_{Qt}^{(8)}(\mu_{\text{EFT}})$) only enter at LO, while the contributions of the mixing effect into $\mathcal{C}_{HG}(\mu_{\text{EFT}})$ or $\mathcal{C}_{tH}(\mu_{\text{EFT}})$ are considered at NLO QCD, and are thus enhanced by a $K$-factor of approximately 2.

In Fig. 3 we investigate the effects of the 4-top operators, setting individually $\mathcal{C}_{Qt}^{(1)}(\mu_0) = 1$ TeV$^{-2}$ or $\mathcal{C}_{Qt}^{(8)}(\mu_0) = 1$ TeV$^{-2}$. Similar to the case of $\mathcal{C}_{tG}$ shown in Fig. 2, $\mathcal{C}_{Qt}^{(1)}$ and $\mathcal{C}_{Qt}^{(8)}$ contribute to a mixing into other Wilson coefficients through RGE evolution, leading to non-vanishing values of all Wilson coefficients depicted in the upper panels.[4] Despite the particular shapes of the distributions, the qualitative observations considering the peculiarities of the running effects are similar to the case of $\mathcal{C}_{tG}$ described above.

---

[4]Note that in the case of $\mathcal{C}_{Qt}^{(1)}(\mu_0)$, the values of $\mathcal{C}_{Qt}^{(1)}(\mu_{\text{EFT}})$ for the variation of $\mu_{\text{EFT}}$ by a factor of two around the central scale do not enclose the values at the central scale for small deviations of $\mu_{\text{EFT}}$ around $\mu_0$. The reason is the lack of the diagonal term for $\mathcal{C}_{Qt}^{(1)}$ in the anomalous dimension, shown in Eq. (2.13), which results in a quadratic dependence, i.e. $\mathcal{O}\left(\left(\alpha_s(\mu_0)\log(\frac{\mu_{\text{EFT}}^2}{\mu_0^2})\right)^2\right)$, of the solution for $\mathcal{C}_{Qt}^{(1)}(\mu_{\text{EFT}})$ when expanded around $\mu_0$.

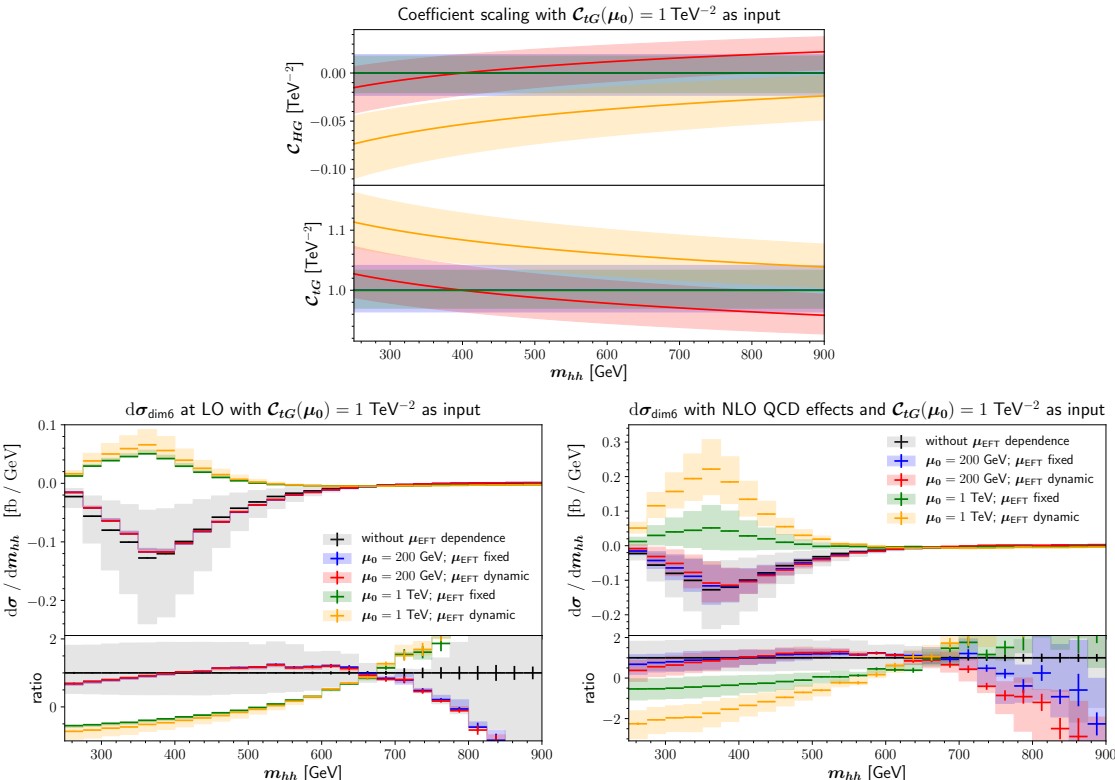

**Figure 2**: Running values of the Wilson coefficients and differential results for $\mathrm{d}\sigma_{\mathrm{dim6}}$ as a function of $m_{hh}$ with different modes of EFT running for $\mathcal{C}_{tG}(\mu_0) = 1$ TeV$^{-2}$ as input. Upper: values of the Wilson coefficients, lower left: $m_{hh}$-distributions for $\mathrm{d}\sigma_{\mathrm{dim6}}$ at LO, lower right: $m_{hh}$-distributions including NLO QCD corrections to the contribution of RGE induced leading Wilson coefficients. Again, the grey bands denote 3-point scale variations of $\mu_R$ and $\mu_F$, while the coloured bands denote variations of $\mu_{\mathrm{EFT}}$ by a factor of two around its central value.

In order to better estimate the impact of the different settings for the running on the full $m_{hh}$ distribution, we present in Fig. 4 the effect of individual SMEFT coefficients $\mathcal{C}_i(\mu_0)$ including the full SM contributions (truncation option (a): $\mathrm{d}\sigma_{\mathrm{SM}} + \mathrm{d}\sigma_{\mathrm{dim6}}$), using the central scales $\mu_R = \mu_F = \mu_{\mathrm{EFT}} = \frac{m_{hh}}{2}$. The input values for the coefficients are oriented at current constraints given in Ref. [55] for $\mathcal{C}_{Qt}^{(1)}$ and $\mathcal{C}_{Qt}^{(8)}$ at $\mathcal{O}(\Lambda^{-2})$, and the ranges resulting from marginalised fits of Ref. [52] for the other coefficients. The SM distribution is depicted including a 3-point scale variation. The distributions with $\mu_0 = 200$ GeV (red) coincide well with the reference distributions (blue) for the leading coefficients; for the subleading coefficients we also find compatibility between blue

and red, except for the threshold region in the cases of $(\mathcal{C}_{tG}, \mathcal{C}_{Qt}^{(1)}, \mathcal{C}_{Qt}^{(8)})$ and the tails of $(\mathcal{C}_{Qt}^{(1)}, \mathcal{C}_{Qt}^{(8)})$. Note that the bands shown in Fig. 4, except for the grey band, denote the variation of the Wilson coefficients within the constraints given on top of the figures, and not the scale variations. If we included scale variations for the SMEFT reference distribution (blue, "without $\mu_{\mathrm{EFT}}$ dependence") the observed differences between the reference and the $\mu_0 = 200\,\mathrm{GeV}$ curves would mostly be within the associated scale uncertainty. Choosing an input scale of $\mu_0 = 1\,\mathrm{TeV}$ (orange), there are observable differences to the case of $\mu_0 = 200\,\mathrm{GeV}$, which are particularly large for $\mathcal{C}_{tG}$, $\mathcal{C}_{Qt}^{(1)}$ and $\mathcal{C}_{Qt}^{(8)}$. Overall, deviations from the SM within current constraints can be large for the leading coefficients and noticeable beyond the SM scale uncertainty for $\mathcal{C}_{tG}$ (both input scales) and $\mathcal{C}_{Qt}^{(1)}$, $\mathcal{C}_{Qt}^{(8)}$ (only for $\mu_0 = 1\,\mathrm{TeV}$).

In the following, we demonstrate the consequences of running effects for a shape benchmark scenario discussed in Refs. [41, 56], investigating the impact on the benchmark scenario 6 specified by Table 1. The distributions for truncation options (a) and

| benchmark | $\mathcal{C}_{H\Box}$ | $\mathcal{C}_H$ | $\mathcal{C}_{tH}(\mu_0)$ | $\mathcal{C}_{HG}(\mu_0)$ |
|---|---|---|---|---|
| SM | 0 | 0 | 0 | 0 |
| 6 | $0.561\,\mathrm{TeV}^{-2}$ | $3.80\,\mathrm{TeV}^{-2}$ | $2.20\,\mathrm{TeV}^{-2}$ | $0.0387\,\mathrm{TeV}^{-2}$ |

**Table 1**: Definition of benchmark scenario 6, considered here in terms of SMEFT Wilson coefficients. Coefficients not listed here are set to 0. Benchmark point 6 refers to the set given in Refs. [41, 56], which is an updated version of Ref. [57]. The benchmark points were originally derived in HEFT, where benchmark point 6 corresponds to $c_{hhh} = -0.684$, $c_{tth} = 0.9$, $c_{tthh} = -\frac{1}{6}$, $c_{ggh} = 0.5$, $c_{gghh} = 0.25$. $C_{HG}$ is determined using $\alpha_s(m_Z) = 0.118$ in the translation between HEFT and SMEFT coefficients.

(b) are depicted in Fig. 5. Since the benchmark scenarios have been originally derived for HEFT, a coefficient translation from HEFT to SMEFT only involves leading Wilson coefficients which do not lead to a mixing in the implemented RGE evolution. Therefore, the qualitative observations made from Fig. 1 apply and the reference distribution is well compatible with the ones including running effects with input scale $\mu_0 = 200\,\mathrm{GeV}$.

We would like to stress that the comparison between the different input scales presented so far refers to different physics scenarios. This should be contrasted to a comparison of equivalent settings of Wilson coefficients defined at different input scales, which we investigate in the following. To this aim we define $\mathcal{C}_{tG}(\mu_0) = 1\,\mathrm{TeV}^{-2}$ at $\mu_0 = 200\,\mathrm{GeV}$ and calculate the values of the Wilson coefficients at $\mu_1 = 1\,\mathrm{TeV}$ using

the contributions to the running described in Sec. 2. Subsequently, the inputs using the derived coefficient values entering at $\mu_1 = 1$ TeV and the original setting of $\mathcal{C}_{tG}(\mu_0) = 1$ TeV$^{-2}$ at $\mu_0 = 200$ GeV are used for the comparison of the $m_{hh}$ distributions shown in in Fig. 6. We observe that the distributions with dynamical $\mu_{\mathrm{EFT}}$ are very well compatible with each other, in contrast to the case shown in the lower panels of Fig. 2. However, comparing the different fixed scale settings for $\mu_{\mathrm{EFT}}$, a sizeable difference is visible. Moreover, considering that the scale evolution of the Wilson coefficients is not available at arbitrary precision, but often only accounts for leading logarithmic RGE terms, the EFT input scale $\mu_0$ should be chosen carefully. We expect a good choice for $\mu_0$ to be given by a value that avoids large logarithmic contributions from the running in the energy range which is optimal to derive constraints from a given observable. On the other hand, this is process or observable dependent and therefore may not be viable for global fits. Nevertheless, coefficients determined at different scales can be compared using external tools for the RGE evolution, which in principle can be provided at better logarithmic accuracy without introducing more computational complexity at runtime of the Monte Carlo evaluation.

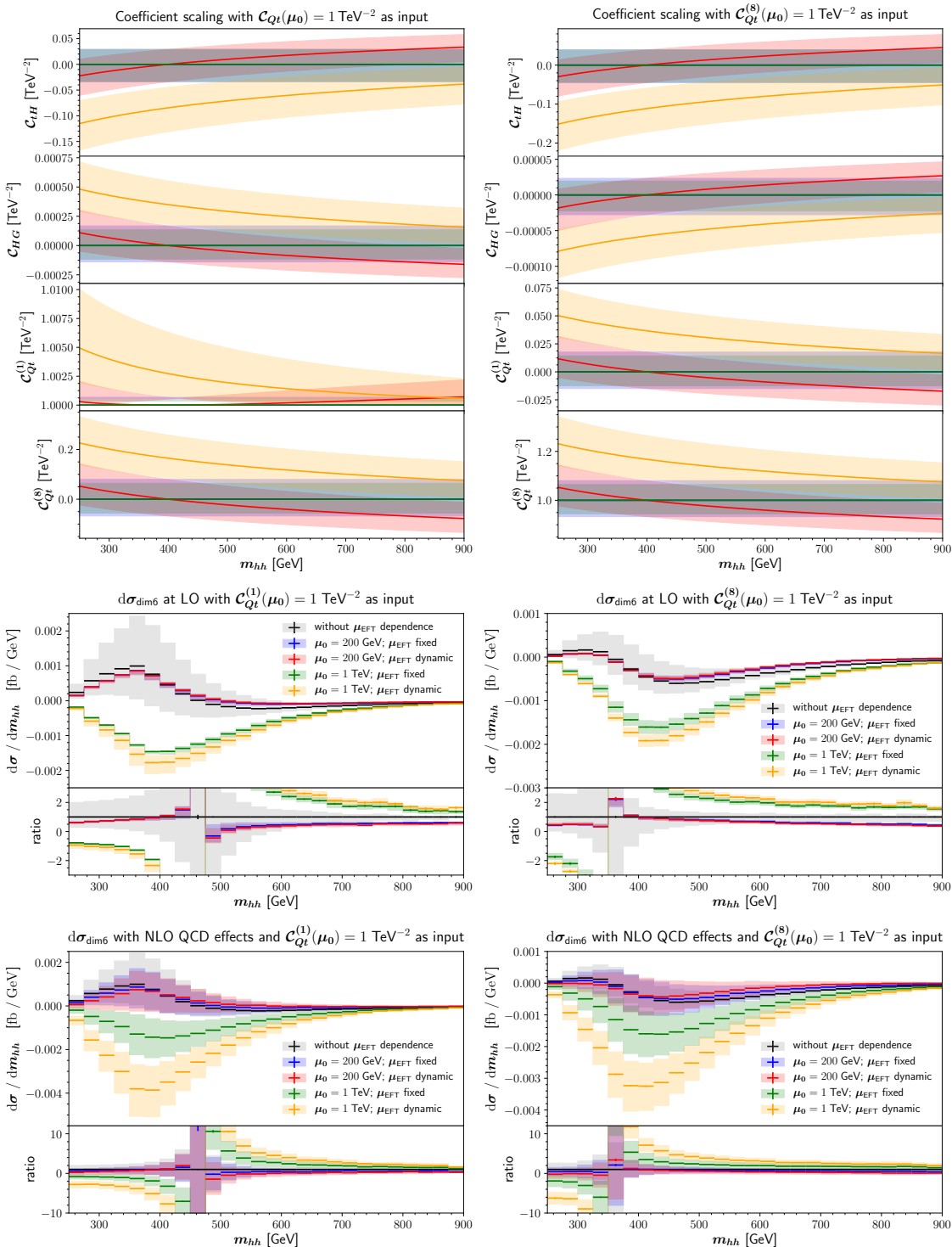

**Figure 3**: Running of the Wilson coefficients and differential cross sections for $\mathrm{d}\sigma_{\mathrm{dim6}}$ as a function of $m_{hh}$ with different modes of EFT running for left: $\mathcal{C}_{Qt}^{(1)}(\mu_0) = 1\ \mathrm{TeV}^{-2}$ as input, and right: $\mathcal{C}_{Qt}^{(8)}(\mu_0) = 1\ \mathrm{TeV}^{-2}$ as input. Upper: values of the Wilson coefficients, middle: $m_{hh}$-distributions at LO, lower: $m_{hh}$-distributions including NLO QCD corrections to the contribution of RGE induced leading Wilson coefficients.

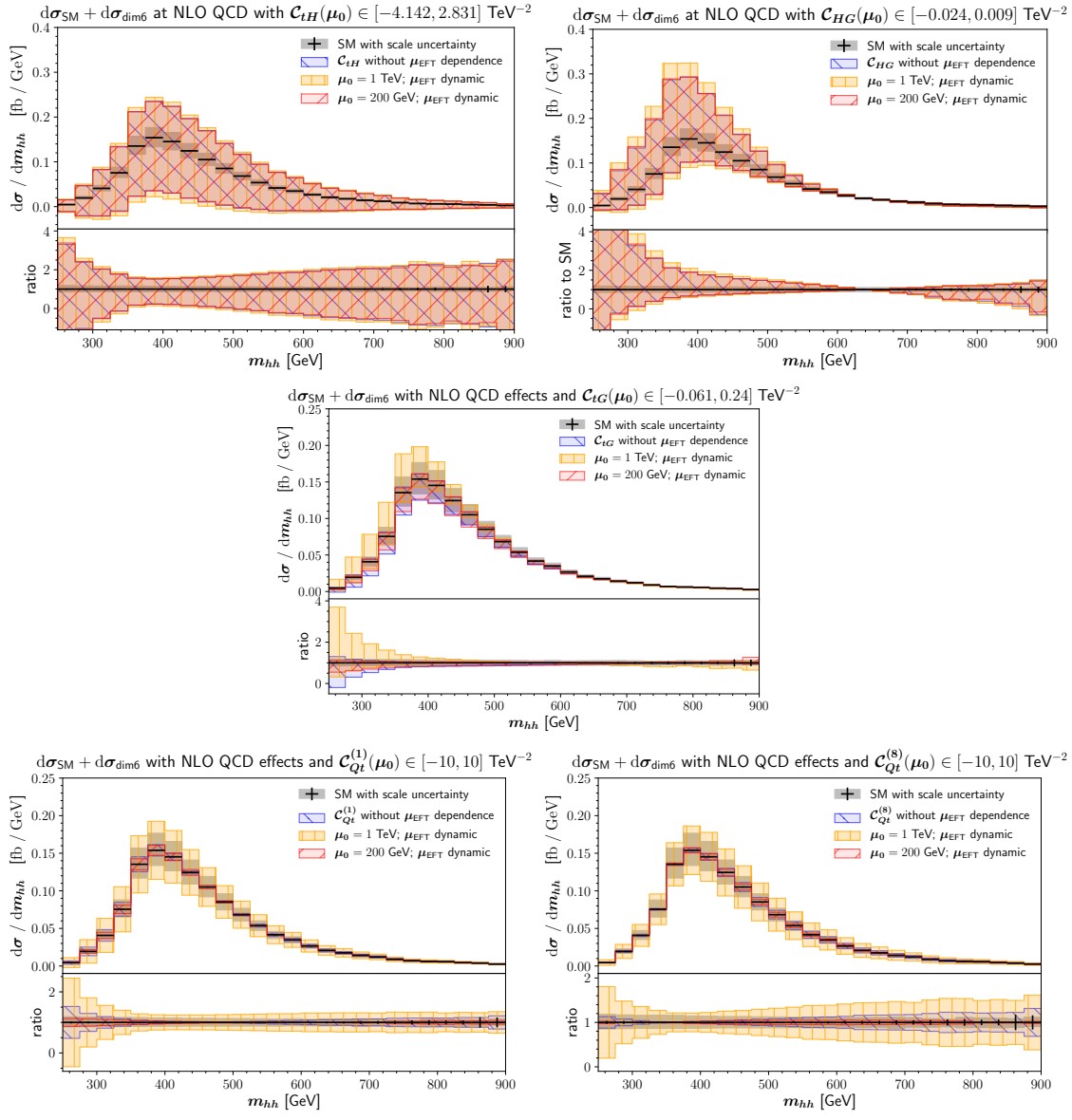

**Figure 4**: $m_{hh}$-distributions demonstrating the impact of individual Wilson coefficient variations on the SM $m_{hh}$-distribution for truncation option (a). The ranges for $\mathcal{C}_{Qt}^{(1)}$ and $\mathcal{C}_{Qt}^{(8)}$ are oriented at the constraints given in Ref. [55], the ranges for the other coefficients are oriented at $\mathcal{O}(\Lambda^{-2})$ marginalised fits of Ref. [52].

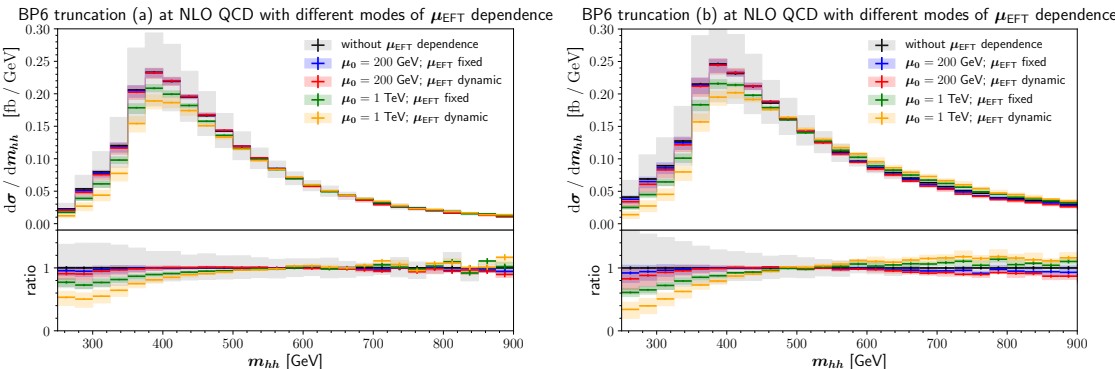

**Figure 5**: NLO QCD distributions of benchmark 6 with different settings for the $\mu_{\text{EFT}}$ scale dependence. Left: truncation option (a), right: truncation option (b). By construction only the Wilson coefficients of the leading SMEFT contribution enter.

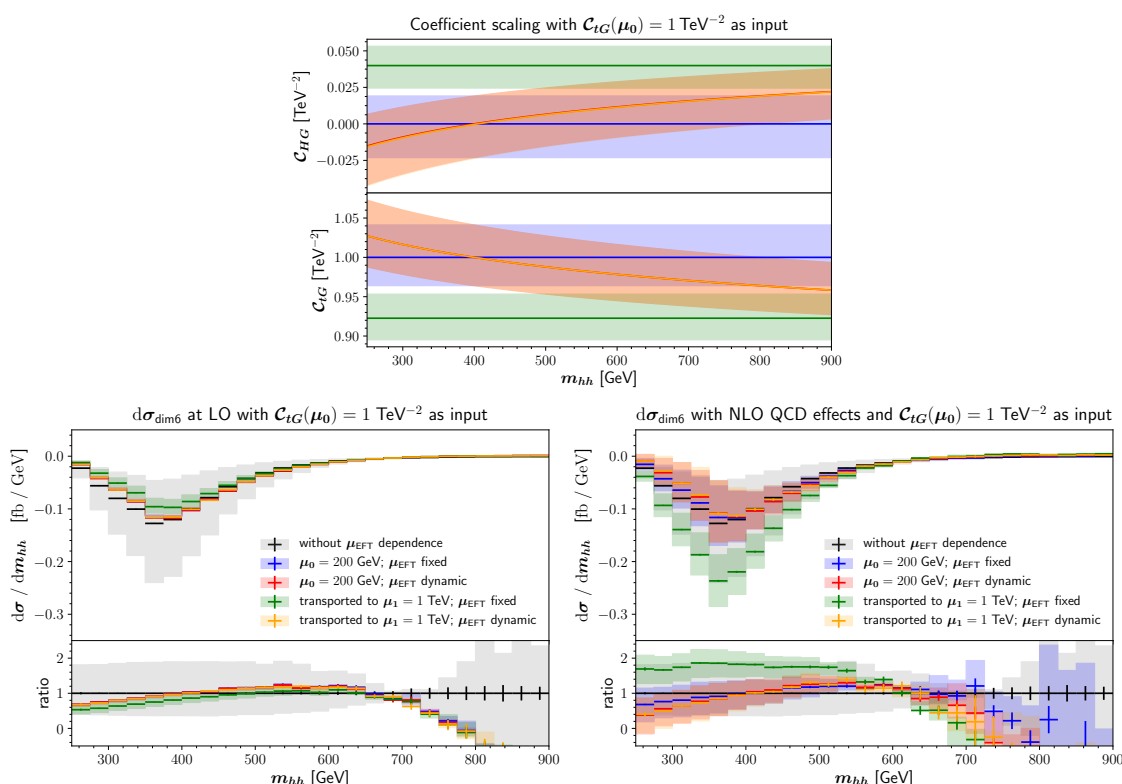

**Figure 6**: Running values of the Wilson coefficients and differential cross section of the linear interference as a function of $m_{hh}$ with different modes of EFT running which are derived from the scenario $\mathcal{C}_{tG}(\mu_0 = 200\,\text{GeV}) = 1\,\text{TeV}^{-2}$. In contrast to Fig. 2, the runs with input at $\mu_1 = 1\,\text{TeV}$ are obtained by first estimating the corresponding physical configuration of the coefficients by running from $\mu_0$ to $\mu_1$. Upper: values of the Wilson coefficients, lower left: $m_{hh}$-distributions at LO, lower right: $m_{hh}$-distributions including NLO QCD corrections to the contribution of RGE induced leading Wilson coefficients.

# 5   Conclusions

We have investigated how the renormalisation group running of Wilson coefficients in SMEFT impact Higgs boson pair production in gluon fusion. After having motivated the selection of relevant contributions to the RGE of the Wilson coefficients, we derived an analytic solution of the scale evolution for the Wilson coefficients that incorporates approximate NLL QCD effects for the leading coefficients. The resulting expressions are added as new features to the public code ggHH_SMEFT [41, 44] which is implemented in the framework of the Powheg-Box-V2 and contains NLO QCD corrections in the SM. We have described the usage of the additional settings in detail.

The effects of the scale dependence of the Wilson coefficients on the process $gg \rightarrow hh$ have been investigated considering both, the running of individual Wilson coefficients as well as $m_{hh}$-distributions where different input choices have been compared. We summarise the main observations in the following.

Choices of the EFT input scale with the same nominal value of the Wilson coefficients but different settings for the running can lead to a drastic change of the effect on the $m_{hh}$ distribution. Therefore, it is important to fully specify the choices made in physical predictions and measurements, in particular the choice for the EFT input scale $\mu_0$.

While the input scale $\mu_0$ can in principle be freely chosen, a convenient choice of $\mu_0$ suppresses the logarithms of the running in the energy range which is optimal to derive constraints for the considered Wilson coefficient, thereby minimising the expected effects of missing electroweak and higher order QCD logarithms. This is achieved by choosing a value within the range of the SM renormalisation scale $\mu_R$ in that region. In predictions for $gg \rightarrow hh$ at the LHC, the SM renormalisation scale is typically set to $\mu_R = \frac{m_{hh}}{2}$, since this scale shows good perturbative stability. Therefore, $\mu_0 \sim m_t$ represents a good choice, as the logarithms for the Wilson coefficient evolution will be suppressed near the top-quark-pair production threshold, which is where the $m_{hh}$-distribution is peaking. Of course, results obtained for different choices for $\mu_0$ can a posteriori be compared by applying external tools for the RGE evolution, which could even take into account electroweak mixing and higher logarithmic accuracy if available.

Formally, calculations in SMEFT for different choices of an EFT basis are equivalent up to a given order in the canonical counting, which includes the freedom to rescale Wilson coefficients by SM couplings. However, different conventions for such rescalings affect the dependence on the SM renormalisation scale $\mu_R$ and on the EFT renormalisation scale $\mu_{\mathrm{EFT}}$ differently. Therefore, the development of a procedure to consistently assess combined SM and EFT scale uncertainties is highly desirable.

## Acknowledgements

We would like to thank Stephen Jones, Matthias Kerner and Ludovic Scyboz for collaboration related to the $ggHH$@NLO project and Stefano di Noi, Ramona Gröber, Michael Spira and Marco Vitti for useful discussions. This research was supported by the Deutsche Forschungsgemeinschaft (DFG, German Research Foundation) under grant 396021762 - TRR 257.

## A  Power counting and the renormalisation group equations

The selection of LL QCD terms of the SMEFT RGE follows the classification in terms of perturbative weight developed in Ref. [58] which employs a generalised version of the naive dimensional analysis (NDA) [59]. The NDA formula for the normalisation of an interaction term in an EFT can be cast in the form [3, 60, 61]

$$\frac{\Lambda^4}{16\pi^2}\left(\frac{\partial_\mu}{\Lambda}\right)^{N_\partial}\left(\frac{4\pi A_\mu}{\Lambda}\right)^{N_A}\left(\frac{4\pi\psi}{\Lambda^{3/2}}\right)^{N_\psi}\left(\frac{4\pi\phi}{\Lambda}\right)^{N_\phi}\left(\frac{g}{4\pi}\right)^{N_g}\left(\frac{y}{4\pi}\right)^{N_y}\left(\frac{\lambda}{16\pi^2}\right)^{N_\lambda}\ , \quad \text{(A.1)}$$

with $N_\partial$ derivatives, $N_A$ gauge fields $A_\mu$, $N_\psi$ spinors $\psi$, $N_\phi$ Higgs fields $\phi$, $N_g$ gauge couplings $g$, $N_y$ Yukawa couplings $y$ and $N_\lambda$ Higgs self-interaction couplings $\lambda$. This NDA normalisation is derived from topological relations and the requirement of an absence of fine tuning and therefore provides an estimate of the relevance of the term up to an $\mathcal{O}(1)$ coefficient. Ref. [62] demonstrated that the counting of generalised chiral dimensions [63] $d_\chi$ can be equivalently used to determine the normalisation, if the assumptions about the underlying dynamics in terms of weak couplings coincide. The minimal assumption resulting in our selection is based on the extraction of a gauge coupling for each field strength tensor and of a Yukawa parameter for a single chirality flipping fermion bilinear in the operator. These conditions correspond to the expectation of fundamental gauge fields and a suppression of chirality flipping terms mediated by the Yukawa couplings; they simultaneously lead to the restoration of a $Z_2$ symmetry which is present in the SM [58].

To clarify the procedure, let us consider the power counting of the terms in Ref. [13] involving QCD mixing between the coefficients $\mathcal{C}_{tH}$, $\mathcal{C}_{HG}$ and $\mathcal{C}_{tG}$ in addition to the

mixing terms of $\mathcal{C}_{tG}$, $\mathcal{C}_{Qt}^{(1)}$ and $\mathcal{C}_{Qt}^{(8)}$ into $\mathcal{C}_{HG}$ of Eq. (2.12)

$$\mu\frac{\mathrm{d}\mathcal{C}_{tH}}{\mathrm{d}\mu} \sim \mathcal{O}\left(\frac{g_s^2}{16\pi^2}\right)\mathcal{C}_{tH} + \mathcal{O}\left(\frac{y_t g_s^2}{16\pi^2}\right)\mathcal{C}_{HG} + \mathcal{O}\left(\frac{y_t^2 g_s}{16\pi^2}\right)\mathcal{C}_{tG}$$
$$+ \mathcal{O}\left(\frac{y_t \lambda}{16\pi^2}, \frac{y_t^3}{16\pi^2}\right)(\mathcal{C}_{Qt}^{(1)} + c_F \mathcal{C}_{Qt}^{(8)}) ,$$
$$\mu\frac{\mathrm{d}\mathcal{C}_{HG}}{\mathrm{d}\mu} \sim \mathcal{O}\left(\frac{g_s^2}{16\pi^2}\right)\mathcal{C}_{HG} + \mathcal{O}\left(\frac{g_s y_t}{16\pi^2}\right)\mathcal{C}_{tG} , \tag{A.2}$$
$$+ \mathcal{O}\left(\frac{y_t^2 g_s^2}{(16\pi^2)^2}\right)(\mathcal{C}_{Qt}^{(1)} + (c_F - \frac{c_A}{2})\mathcal{C}_{Qt}^{(8)}) ,$$
$$\mu\frac{\mathrm{d}\mathcal{C}_{tG}}{\mathrm{d}\mu} \sim \mathcal{O}\left(\frac{g_s^2}{16\pi^2}\right)\mathcal{C}_{tG} + \mathcal{O}\left(\frac{y_t g_s}{16\pi^2}\right)\mathcal{C}_{HG} .$$

We normalise the terms of the Warsaw basis according to the classification of chiral dimensions which in terms of the Wilson coefficients $\mathcal{C}_i$ can be written as [45]

$$\mathcal{C}_i = g_s^{N_G} y_t^{N_{\bar{Q}\tilde{\phi}t}}\left(\frac{1}{16\pi^2}\right)^{(d_\chi - 4)/2}\frac{\tilde{C}_i}{\Lambda^2} , \tag{A.3}$$

where $N_G$ is the number of gluon field strength tensors of the operator and $N_{\bar{Q}\tilde{\phi}t}$ determines whether there is a single chirality flipping fermion bilinear. The coefficients of the normalised Lagrangian on the right-hand side of Eq. (A.3) are expected to be $\tilde{C}_i \sim \mathcal{O}(1)$ following the considerations presented at the beginning of this Appendix. For the relevant Wilson coefficients we explicitly have

$$\mathcal{C}_{tH} = y_t(16\pi^2)\frac{\tilde{C}_{tH}}{\Lambda^2} ,$$
$$\mathcal{C}_{HG} = g_s^2\frac{\tilde{C}_{HG}}{\Lambda^2} ,$$
$$\mathcal{C}_{tG} = g_s y_t\frac{\tilde{C}_{tG}}{\Lambda^2} , \tag{A.4}$$
$$\mathcal{C}_{Qt}^{(1/8)} = (16\pi^2)\frac{\tilde{C}_{Qt}^{(1/8)}}{\Lambda^2} ,$$

which leads to

$$
\mu\frac{\mathrm{d}\tilde{C}_{tH}}{\mathrm{d}\mu} \sim \mathcal{O}\left(\frac{g_s^2}{16\pi^2}\right)\tilde{C}_{tH} + \mathcal{O}\left(\frac{g_s^4}{(16\pi^2)^2}\right)\tilde{C}_{HG} + \mathcal{O}\left(\frac{y_t^2 g_s^2}{(16\pi^2)^2}\right)\tilde{C}_{tG}
$$
$$
+ \mathcal{O}\left(\frac{\lambda}{16\pi^2}, \frac{y_t^2}{16\pi^2}\right)(\tilde{C}_{Qt}^{(1)} + c_F \tilde{C}_{Qt}^{(8)}) \,,
$$
$$
\mu\frac{\mathrm{d}\tilde{C}_{HG}}{\mathrm{d}\mu} \sim \mathcal{O}\left(\frac{g_s^2}{16\pi^2}\right)\tilde{C}_{HG} + \mathcal{O}\left(\frac{y_t^2}{16\pi^2}\right)\tilde{C}_{tG} \tag{A.5}
$$
$$
+ \mathcal{O}\left(\frac{y_t^2}{16\pi^2}\right)(\tilde{C}_{Qt}^{(1)} + (c_F - \frac{c_A}{2})\tilde{C}_{Qt}^{(8)}) \,,
$$
$$
\mu\frac{\mathrm{d}\tilde{C}_{tG}}{\mathrm{d}\mu} \sim \mathcal{O}\left(\frac{g_s^2}{16\pi^2}\right)\tilde{C}_{tG} + \mathcal{O}\left(\frac{g_s^2}{16\pi^2}\right)\tilde{C}_{HG} \,.
$$

This provides the form of the RGE in which we identify the relevant terms:

- Applying the NDA normalisation to the selected coefficients, the one-loop QCD contributions in the RGE reduce to QCD corrections diagonal in the coefficients for $\tilde{C}_{tH}$, $\tilde{C}_{HG}$ and $\tilde{C}_{tG}$ up to the mixing of $\tilde{C}_{HG}$ into $\tilde{C}_{tG}$. The mixing of $\tilde{C}_{HG}$ and $\tilde{C}_{tG}$ into $\tilde{C}_{tH}$ are of higher order and suppressed by $g_s^2(16\pi^2)^{-1}$ and $y_t^2(16\pi^2)^{-1}$, respectively.

- The mixing of $\tilde{C}_{tG}$ into $\tilde{C}_{HG}$ and $\tilde{C}_{Qt}^{(1)}$, $\tilde{C}_{Qt}^{(8)}$ into $\tilde{C}_{tH}$, $\tilde{C}_{HG}$ required by the renormalisation of the subleading contribution are not of QCD origin, but describe one-loop effects. In particular, the mixing of $\mathcal{C}_{Qt}^{(1)}$, $\mathcal{C}_{Qt}^{(8)}$ into $\mathcal{C}_{HG}$ originating from two-loop diagrams is of the same order as the mixing of $\tilde{C}_{tG}$ into $\tilde{C}_{HG}$ in this power counting approach. Note that we do not consider any terms at $\mathcal{O}\left(\frac{y_t^2}{16\pi^2}\right)$ and $\mathcal{O}\left(\frac{\lambda}{16\pi^2}\right)$, but only include those induced by the renormalisation in Eq. (2.10).

- Considering a weakly coupling, renormalisable UV completion on top of the assumptions made in the beginning does not change the picture. Including the weak couplings of the UV theory entering a matching calculation in the counting of $d_\chi$ for the expansion in Eq. (A.3) results in a tree-loop classification, see Ref. [45]. This changes the overall normalisation of the coefficients in Eq. (A.4) by a factor of $(16\pi^2)^{-1}$, but the hierarchy between the selected coefficients remains intact. In particular, the weights determined by Eq. (A.5) are still valid.

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
