# Peer review of "Renormalisation group effects in SMEFT for di-Higgs production"

_SciPost Physics Core_

## Round 1 · Referee Report · Anonymous (Referee 1) · 2025-1-7

Report

In this paper the authors study renormalization-group (RG) effects in the SMEFT for di-Higgs production at the LHC, supplementing the NLO SMEFT QCD calculation from Ref. [41] with the running of Wilson coefficients within certain approximations spelled out in the paper.

The di-Higgs production process is of considerable importance to the goal of deciphering the shape of the Higgs potential, and given that no new physics has been observed at the LHC, the SMEFT has become the default tool for investigating BSM effects in a model-independent way. The current paper studies important aspects of the NLO SMEFT computation of this process,
and is therefore of considerable topical interest and the results are publishable in SciPost.

That said, one might argue that the RG-running studied in the present paper is a standard part of an NLO EFT calculation and would have been better presented along with the NLO SMEFT QCD calculation in Ref [41]. While I understand that there can be reasons for splitting the work into two pieces like this, it puts a burden on the reader to assimilate two papers at once, and I think some changes to the text could make this an easier task.

In particular, my understanding of the RG evolution considered in the present work is that it allows to vary the scale $\mu_{\rm EFT}$ appearing both in the renormalised NLO matrix elements of the SMEFT operators, and in the Wilson coefficients $C_i(\mu_{\rm EFT})$. The dependence on $\mu_{\rm EFT}$ then cancels between the Wilson coefficients and the matrix elements, up to terms of NNLO and higher. In order to avoid large logarithms in the matrix elements one chooses $\mu_{\rm EFT}\sim \mu_R \sim E$, where $E\sim m_h$ or $m_{hh}$ is the characteristic energy scale of the process. The Wilson coefficients $C_i(\mu_{\rm EFT})$ are obtained from those at a fixed reference value $\mu_0$ through the exact solutions to RG equations, even though in practice the authors make the reasonable choice $\mu_0 \sim E$ so that no large logarithms are being resummed (in contrast to the case where $\mu_0\sim \Lambda$, with $E/\Lambda \ll 1$ in the EFT counting).

Given the above picture, which is standard in the NLO SMEFT literature, I would ask that the authors address the following points:

1) As written, the default scheme on pg. 9 of the text and used as the reference scheme in Section 4 seems to set $\mu_{\rm EFT}=\mu_R$ in the NLO SMEFT matrix elements, with $\mu_R$ varied, while in the Wilson coefficients $C_i=C_i(\mu_0)$ is kept static (i.e. do not depend on $\mu_R$). If that were the case, then the cross section would have explicit $\mu_R$ dependence at NLO, instead of at NNLO and higher, and the scale variation of the cross section with $\mu_R$ would not be a reliable estimate of missing NNLO terms, as is usually assumed to be the case. If this is so, the authors should point it out in the text; if instead I misunderstand the scheme, the authors should change the wording so that the wrong interpretation is not possible.

2) When performing an NLO calculation in SMEFT (as in the SM), one would normally like to see that 1) scale uncertainties decrease at NLO compared to LO, and 2) the NLO results lie within the uncertainty band of LO (at least for Wilson coefficients connected to the LO result through RG running). The presentation in Section 4 makes it hard to see quantitatively whether this happens, as LO and NLO are never shown on the same plot. To make clear the behavior of the perturbative series, I would suggest that the authors present numbers for the total cross section at LO and NLO in SMEFT, including scale variations in at least one benchmark scenario.

In addition to the points above, I also have a few minor comments:

pg. 3: weakly coupling UV theories $\to$ weakly coupled UV theories

Eq. (2.5): the running of the Yukawa coupling $y_t$ also receives SMEFT QCD contributions -- explain why they are irrelevant.

Eq. (2.11) and elsewhere: I do not see a definition for $v$: what EW input scheme is being used?

pg 5: should reword so that the paragraph doesn't end with "guiding principles:" (one normally doesn't end a paragraph with a colon).

pg 6: It is written ``we employ a decoupling of heavy particles like the top-quark and the Higgs boson from the RGE, as their logarithmic contributions should not be of high impact". However, only logs $\mu_0/\mu_{\rm EFT}$ are being resummed by RG evolution, and in practice
$\mu_0\sim \mu_{\rm EFT}$ is not far from $m_t$ or $m_h$, so are the logarithms related to decoupling really smaller than others being considered?

pg 8: desireable $\to$ desirable

After these points are cleared up, I would recommend the paper for publication.

Recommendation

Ask for minor revision

---

## Round 1 · Referee Report · Anonymous (Referee 2) · 2025-1-14

Report

This is a timely paper that is relevant to the current, high-profile exploration of di-Higgs final states at the LHC. The paper is clear, well-written, and provides a concise analysis of RGE effects. Results are also provided as a Monte Carlo tool that will assist the experimental and phenomenology communities with further steps and investigations. I recommend publication in SciPost.

Recommendation

Publish (meets expectations and criteria for this Journal)

---

## Editorial Decision

resubmitted